# The EPH/Ephrin System in Pancreatic Ductal Adenocarcinoma (PDAC): From Pathogenesis to Treatment

**DOI:** 10.3390/ijms24033015

**Published:** 2023-02-03

**Authors:** Stavros P. Papadakos, Nikolaos Dedes, Nikolina Gkolemi, Nikolaos Machairas, Stamatios Theocharis

**Affiliations:** 1First Department of Pathology, Medical School, National and Kapodistrian University of Athens, 11527 Athens, Greece; 2Second Department of Propaedeutic Surgery, National and Kapodistrian University of Athens, Laikon General Hospital, 11527 Athens, Greece

**Keywords:** EPH/eprin, signaling pathway, PDAC, pancreatic cancer, immunotherapy

## Abstract

Pancreatic ductal adenocarcinoma (PDAC) is a major concern for health care systems worldwide, since its mortality remains unaltered despite the surge in cutting-edge science. The EPH/ephrin signaling system was first investigated in the 1980s. EPH/ephrins have been shown to exert bidirectional signaling and cell-to-cell communication, influencing cellular morphology, adhesion, migration and invasion. Recent studies have highlighted the critical role of the EPH/ephrin system in various physiologic processes, including cellular proliferation, survival, synaptic plasticity and angiogenesis. Thus, it has become evident that the EPH/ephrin signaling system may have compelling effects on cell homeostasis that contribute to carcinogenesis. In particular, the EPH/ephrins have an impact on pancreatic morphogenesis and development, whereas several EPHs and ephrins are altered in PDAC. Several clinical and preclinical studies have attempted to elucidate the effects of the EPH/ephrin pathway, with multilayered effects on PDAC development. These studies have highlighted its highly promising role in the diagnosis, prognosis and therapeutic management of PDAC. The aim of this review is to explore the obscure aspects of the EPH/ephrin system concerning the development, physiology and homeostasis of the pancreas.

## 1. Introduction

Pancreatic ductal adenocarcinoma (PDAC) is the most common histological type of pancreatic cancer and accounts for approximately 90% of pancreatic malignancies [1]. The incidence of the disease has progressively increased on a global scale, emerging as the seventh leading cause of cancer-related deaths worldwide [2]. Early detection remains a major challenge, since PDAC becomes symptomatic at advanced stages. Patients typically present with locally advanced disease and/or with distant metastases, thus rendering them inoperable. Indicatively, patients with early-stage PDAC (stage IA) comprise only 1.8% of all cases [3]. 

Whole-genome sequencing has contributed significantly to the deciphering of the molecular landscape in PDAC. *KRAS*, *TP53*, *CDKN2A* and *SMAD4* are the four most frequently mutated genes in PDAC [4]. The *BRCA*, *APOBEC* and *KDM6A* genes are involved in the stabilization and remodeling of chromatin in tumor cells. The frequency of *BRCA* mutations is approximately 5.9–7.2%, and a plethora of emerging data suggests that these patients would experience clinical benefits with PARP inhibitors [5]. It is apparent that the aforementioned molecules present as potential therapeutic targets for the precise medical management of PDAC. Nevertheless, targeted therapies have not achieved significant improvement in the overall survival (OS) of patients with surgically unresectable diseases [5]. Immunotherapy for PDAC is a rapidly developing research field with poor clinical outcomes so far [6]. Various immune checkpoint inhibitors (anti-CTLA-4, anti-PD-1) have been studied either as monotherapies, as dual immune checkpoint inhibitors or combined with chemotherapeutic regimens [6]. Other immunotherapeutic modalities, such as immune vaccines [7] and tumor stroma modulators [8], are still undergoing preclinical studies. 

The PDAC’s TME is characterized by its immunosuppressive nature, which is the end result of several cellular and molecular mechanisms [9]. Granulocyte–macrophage colony-stimulating factor (GM-CSF), CCL2 [10], CSF18 and BAG3 [11] orchestrate the infiltration, maturation and differentiation of myeloid cells into myeloid-derived suppressor cells (MDSCs) and tumor-associated macrophages (TAMs), respectively. MDSCs and TAMs suppress the anti-tumor cellular responses, promoting an epidermal growth factor receptor (EGFR)/mitogen-activated protein kinase (MAPK)-driven upregulation of programmed death-ligand 1 (PDL1) [12]. The selective inhibition of myeloid cells targeting the CSF1/CSF1R signaling sensitized PDAC in immune checkpoint inhibition (ICI) [12]. Concerning the T cell populations, PDAC is considered a “cold” tumor due to a lack of effector CD8+ T cells [9] and a relative predominance of Foxp3+ regulatory T cells (Tregs). Tregs exert their immunosuppressive effects on PDAC secreting transforming growth factor b (TGFb) and interleukin (IL)-10, which have well-described immunosuppressive functions [9]. Tregs, also downregulates the activation of CD8+ T cells and the generation of interferon gamma [9]. Thus, a relative increase/decrease in Tregs/CD8+ T cells is documented during the progression of premalignant pancreatic intraepithelial neoplasias and intraductal papillary-mucinous neoplasms into PDAC [13]. CXCL13 [14] and Bruton tyrosine kinase (BTK) [15] guide the aggregation of B-cells into TME and hypoxia-inducible factor 1a (HIF-1a) and exert the opposite effects [16]. While evidence has suggested that B cell infiltration promotes PDAC progression [16], currently, its clinical significance has not been exhaustively elucidated. Despite the fact that CXCL12-CXCR4 signaling influences a pancreatic stellate cell (PSCs)-mediated activation of CD8+ T cells and, consequently, sensitization to ICI [17], the above effects are context-specific and more evidence is needed to shed light on the pathogenic mechanisms. Aside from the above-mentioned cellular immunosuppressive population, there are molecular suppressive mechanisms in PDAC’s TME. Focal adhesion kinase (FAK) drives the accumulation of fibrosis in preclinical mouse models and limits CD8+ T-cell augmentation. FAK targeting shrinks the accompanying fibrotic stroma, limits the infiltration of myeloid cells and enhances the response to ICI [18]. In parallel, components of the extracellular matrix, such as hyaluronic acid, increase the interstitial pressure and restrict tumor vasculature-restraining drug delivery [19]. 

Given the fact that the EPH/ephrin signaling system in the pancreas comprises an increasingly studied pathway, the aim of our review was to investigate its implication in the pathogenesis of PDAC as well as its possible contribution to the management of disease, either as biomarker to guide therapeutic decision or as therapeutic target. 

### Molecular Characteristics of the EPH/Ephrin Signaling Pathway

Erythropoietin-producing hepatocellular receptors (EPHs), named for the liver cancer cell line from which they were cloned, represent the largest subfamily of receptor tyrosine kinases (RTK) [20]. Unlike other RTKs, EPHs interact with membrane surface-associated ligands called ephrins. The EPHs and their ligands, the ephrins, are divided into two subclasses (A and B) with five ephrin-A ligands, three ephrin-B ligands, nine EPHA and five EPHB receptors in the human genome [21,22]. The confirmed combinations of EPH/ephrin are the nine EPHA receptors, which bind five ephrin-A ligands, and the five EPHB receptors, which bind three ephrin-Bs. Additionally, EPHA4 and EPHB2 can bind ephrins of a different class. The EPH/ephrin family is present in a multitude of tissues, showing a combinatorial nature as well as dynamically changing expression patterns [20].

The modus operandi of EPHs and ephrins typically includes cell-to-cell adhesion and communication. More specifically, the EPH/ephrin system mediates contact-dependent interaction between the same or different cell types in order to control physiological cell activities during development, such as cell morphology, adhesion, movement, proliferation, survival and differentiation. Interestingly, all the aforementioned physiological functions are cornerstones in the complicated and multifactorial route of carcinogenesis and cancer progression [23]. In particular, angiogenesis and lymphangiogenesis are two of the most significant mechanisms that malignant tumors exploit in order to achieve rapid growth and distant metastatic dissemination [24]. The EPH/ephrin signaling establishes borders between different compartments and helps in the remodeling process by regulating the effects of vascular growth factors [25].

There are two types of interactions between EPHs and their ligands: trans-interaction, where the expression of ephrin and EPH is located in opposing cells; this activates bi-directional signaling, triggering a response in the cytoplasm of the receptor-expressing cell (forward signaling) as well as in the ephrin-expressing cell (called reverse signaling). The second is cis-interaction, in which both EPH and ephrin are expressed in the same cell [22]. In more detail, cis-binding occurs via the N-terminal region of the tyrosine kinase domain (e.g., EPHA4/FGFR1), the tyrosine kinase domain (e.g., EPHA2/Dvl2), the extracellular (e.g., EPHA2/Meltrinβ) or intracellular domain (ephrinB1/RhoGDI1) and through the intercession of intermediate proteins (e.g., EPHB2/ADAM10/E-cadherin) [26]. These less-mentioned interactions play a key role in important cellular processes. Mounting evidence has linked cis-interactions with cell cycle control. One such example is EPHA2, which may interact with members of the EGFR family or cytoplasmic proteins (such as Dvl2 or YAP) and thus lead to cell proliferation in an ephrin-independent, yet Ras/ERK-mediated manner [27,28]. In addition to cell proliferation, cell adhesion and migration may also be influenced by cis interactions. In this context, ephrin-A1/EPHA2 union leads to Src recruitment, which, in turn, engages with intergrin and facilitates cell motility [29]. Finally, cis-interactions are implicated in cell-sensing (the ability of cells to adapt to the topography of their environment). This feature is crucial to the development of the nervous system and axon guidance. Relative to this, activated EPHA4 interacts with alpha2-chimaerin, which leads to RhoA activation and Rac1 inhibition. This cascade eventually results in growth cone collapse [30,31]. Generally, EPH-dependent cellular interactions are involved in cytoskeletal rearrangements, such as the collapse of the cytoskeleton, by exerting their action on the equilibrium between the activation and inactivation of small GTPase. EPH forward signaling leads to cell repulsion, whereas ephrin reverse signaling evokes either cell repulsion or adhesion [32]. Forward signaling is determined by the interplay of EPHs and ephrins with a variety of signaling pathways, e.g., Rho and RasGTPases, phosphoinositide 3-kinase (PI3K), focal adhesion kinases (FAK) and Janus kinase (JAK)-signal transducer, which is an activator of transcription (STAT) [33]. Concerning backward signaling, the signal transmission is conducted through proteins that contain Src Homology 2 (SH2) or PDZ domains, such as Grb4, which bind with ephrins and result in their phosphorylation [21]. The widespread expression of EPH and ephrins in various cell types leads to their participation in many different physiological functions, which are, at the same time, scaffolds in cancer development. The morphology of the EPH/ephrin system is illustrated in Figure 1. 

## 2. The Role of EPH/Ephrin System in the Pancreas

### 2.1. The EPH/Ephrin System in Pancreatic Embryology and Physiology

The EPH/ephrin system has been associated with numerous processes involving the embryologic integration of the pancreatic parenchyma and the positioning of the islets of Langerhans [34], which represent a major endocrine component in the regulation of insulin secretion [35]. Transcriptomic analyses of the main exocrine and endocrine pancreatic cells demonstrated that the sophistication of the system is determined by compounded heterotypic cellular interactions. The EPH/ephrin system, in tandem with 7-Transmembrane receptors (7-TM receptors) and ligands from the TGF-b class, compose the main regulators of the heterotypic synergy among the aforementioned cellular compartments [36]. The alpha and beta endocrine cellular populations are characterized by the overexpression of EFNA5 and EFNB3, while EFNA1 and EFNB2 predominate in the small and large ducts and acinar cells. A more in-depth presentation of those interactions has been given elsewhere, and this goes beyond the scope of our manuscript [36]. Class B of the EPH/ephrins orchestrate the pancreatic morphogenesis. They appear earlier than class A molecules, at embryonic day 12.5, and regulate the alignment of the pancreatic epithelium, branching and lumen formation. The interplay among the epithelium-expressed EPHB2 and EPHB3 and their concomitant ligands in the pancreatic arteries and mesenchyme [35,36,37] mediates the expression of several cell adherence molecules, such as junctional b-catenin and E-cadherin [38]. Roughly, EPHB3 comprises the only EPH/ephrin molecule that is expressed in mesoderm. Its interplay with the endodermal ephrin-B1 guides the formation of the extrahepatic bile duct, the gallbladder and the common bile duct. The EPHB3-EPHB4 interaction contributes to gallbladder formation, the EPHB3-ephrinB2 regulates the development of the gallbladder and common bile duct and the EPHB3-EPHB3 in the endoderm directs the composition of the extrapancreatic duct [39]. Since pancreatic morphogenesis emerges as a summation of consecutive processes, e.g., the arrangement of the epithelium into distinct layers, the periodic loss of apical–basal polarity and epithelial tubule reconstruction, it is highly dependent on EPHB signaling. Contrarily, several findings have suggested that the class A EPH/ephrins comprise a central regulator of insulin secretion. It is well-established that the metabolism of glucose in β-cells stimulates basal insulin secretion [40], while the interactions between β-cells shape insulin secretion in response to glucose [41]. This process is essential to achieving the suppression of insulin secretion during starvation and adequate amounts of insulin during feeding [42]. Presently, it is common knowledge that blood glucose levels regulate insulin secretion, exerting their effects on class A EPH/ephrins. At high glucose levels, the dephosphorylation of EPHA5 by protein tyrosine phosphatases (PTPs) suppresses the EPH forward signaling. The unopposed ephrin-A5 backward signaling results in insulin secretion. On the other hand, at low glucose concentrations, the forward signaling outweighs the reverse signaling, inhibiting the insulin secretion. Insulin secretion could result either from the suppression of EPHA signaling or from the enhancement of ephrin-A reverse signaling [42]. Analogously, in α-cells, the enhancement of EPHA4 forward signaling suppresses the glucagon secretion [43]. All of the above indicate that the shaping of pancreatic morphology and physiology are interconnected and the EPH/ephrin system exerts major influence on their configuration.

### 2.2. The EPH/Ephrin System in PDAC—Preclinical Data

EPHA2 and EPHA4 are the most important targets in the field of PDAC translational research [44]. EPHA2 has attracted the attention of the research community due to its involvement in tumor capillary formation [45]. Despite the fact that the initial attempts to target EPHA2 in order to enhance the specificity of adenoviral vectors were not fruitful [46], the development of EPHA2-specific antibody agonists and ephrinA1 antagonists suppressed tumor growth and metastatic disease, inhibiting angiogenesis in mice with orthotopical transplantation of MiaPaCa2 cells [47]. In their groundbreaking study, Markosyan et al. analytically investigated the role of EPHA2 in PDAC [48]. The CRIPSR-Cas9-mediated generation of Epha2-KO congenic mice by 6419c5 and 6694c2 cell lines, which have low T-cell infiltration levels, exhibited substantially modified immune microenvironment in comparison with the wild-type cell lines. They documented, in *Epha2-KO* tumors, enhanced infiltrates of CD4 + and CD8 + T-cells, with diminished presence of myeloid and myeloid-derived suppressor cells (MDSCs) and unaltered numbers of antigen-presenting cells, such as macrophages and dendritic cells. The therapeutic combination of gemcitabine, nab-paclitaxel, anti-CD40 agonists, anti–CTLA-4 and anti–PD1-1 in *EPHA2-KO* tumors achieved results that were more efficacious than those for *EPHA2*-wild-type tumors, but comparable with those for high T-cell-infiltrating ones. Altogether, the above strongly suggest that EPHA2 exerts modifying properties over the immune tumor microenvironment (TME) [48]. A series of sophisticated experiments unfolded the existence of the EPHA2/TGF-β/PTGS2 pathway. The prostaglandin endoperoxide synthase 2 (PTGS2) gene encodes the cyclooxygenase-2 (COX-2). The tumor-accelerating properties of PTGS2 are owed to its efficacy in activating downstream signaling pathways such as the RAS [49], PI3K/AKT [50] and ERK [51]. The identification of this pathway could offer novel therapeutic avenues in the medical management of PDAC, since the COX-2 inhibition could sensitize PDAC to immunotherapy [52]. Finally, there is an auspicious perspective that the utilization of EPHA2 as a surface marker to increase the sensitivity of exosomal collection and assortment will provide an invaluable source of clinical data [53]. Recent evidence implicated ephrin-A5 in the development of fibrotic stroma [54]. Nakajima et al. documented a significant reduction in collagen density (type I, III and IV collagen) upon exposure to neoadjuvant therapy (NAT). In human-derived PDAC cell cultures, it was evident that ephrin-A5 signaling regulated the expression of several genes implicated in collagen synthesis. Collectively, NAT inhibited the expression of CAFs, shaping the PDAC microenvironment, and indirectly inhibited PDAC cells, reducing the fibrotic stroma through *EFNA5* downregulation [54].

EPHB4/ephrin-B2 is signaling pathway which has been the studied in the most depth with regard to the class B EPH/ephrin system [55]. Initial studies reported that soluble EPHB4 blockers, inhibiting the ephrin-B2 forward signaling in venous endothelial cells and the backward signaling in the arterial endothelium, diminish tumor growth. This became more evident with additional Dll4/Notch inhibition [56]. Aside from its effects on angiogenesis, ephrinB2 signaling influences cellular proliferation and migration, exerting its impact on the cell cycle and epithelial–mesenchymal transition (EMT). In more detail, Zhu et al. demonstrated that the *EFNB2* knockdown upregulates p53, inducing a fixation in the G0/G1 phase and cell cycle arrest. In parallel, an upward trend occurred in E-cadherin expression with a concomitant downregulation of vimentin, which are decidedly suggestive of an influence of ephrin-B2 signaling on cellular invasion through EMT regulation [57]. EPHB4 suggests an attractive cytotoxic target in PDAC. In vivo data from orthotopic xenografts showed enhanced tumor growth retardation with the addition of EPHB4 inhibition in combination with gemcitabine [55]. Furthermore, data associating the EPHB4/ephrinB2 signaling with the modulation of PDAC ΤΜΕ have begun to emerge [58,59]. Radiotherapy (RT) induces immune infiltration, attracting both anti-tumorigenic (effector T-cells and interferon I signaling activation) and pro-tumorigenic (regulatory T-cells (Tregs), tumor-associated macrophages (TAMs) and myeloid-derived suppressor cells (MDSCs) cellular populations [58]. Lennon et al. documented, both in vitro and in vivo, that the inhibition of EPHB4/ephrin-B2 signaling in conjunction with RT shifts the balance towards the anti-tumor responses, reducing PDAC tumor growth and limiting the fibrotic response [59]. This could have major clinical applications in the therapeutic management of PDAC. The above are briefly illustrated in Figure 2.

### 2.3. The EPH/Ephrin System in PDAC—Clinical Data

The significance of the EPH/ephrin system for PDAC became conceivable due to its overexpression in a multitude of studies [48,55,60,61]. EPHA2 is the most clinically relevant member of class A EPH/ephrin signaling. Despite the fact that the earliest references in the literature were restrained regarding its role in PDAC carcinogenesis, only documenting association with patients’ age [60], data concerning its actual impact have begun to emerge [48,61]. Van den Broecket et al. reported that EPHA2 has been overexpressed in PDAC, with unfavorable clinical outcomes [60]. The above finding is in accordance with a clinical study by Nakajima et al., which documented that EPHA2 was expressed in the vast majority of PDAC cases with variable density. EPHA2 was stained principally in the cancer cells and, to a lesser extent, in CAFs. An association with a more invasive tumor phenotype was also documented [53]. Markosyan et al. confirmed via human clinical samples that the EPH/ephrin system is one of the most ubiquitously expressed signaling pathways in T cell noninflamed PDAC, with EPHA2 being the principally expressed gene. The expression of *CD8A*, *CD3*, *PRF1* and *GZMB* mRNA levels exhibits a negative association with EPHA2, which collectively suggests that the EPHA2 possesses immune-modifying properties [48]. 

EPHA2 also displays clinical usefulness as a biomarker; Koshikawa et al. documented an 89.0% sensitivity and 90.0% specificity of soluble EPHA2 fragments in PDAC diagnosis, in opposition with the respective 88.9% and 72.0% of the Ca19-9 [62]. Wei et al. suggested that the combination of serum exosomal EPHA2 with Ca19-9 could potently distinguish early-stage pancreatic cancer (stage I, II) from benign pancreatic disease [63]. The above could reshape the diagnostic management of pancreatic cancer, constituting useful alternatives for population screenings. Finally, monoclonal antibodies against EPHA2 are under clinical investigation without evidences of dose-limiting toxicity or adverse events [64]. 

Regarding the class B EPH/ephrin system, its importance has been also recognized in human clinical studies. The EPHB4 and ephrin-B2 overexpression shape, in conjunction with several other genes, a more malignant clinical phenotype [61], which is partially parallel to the fact that ephrin-B2′s expression correlates with the TNM Classification of Malignant Tumors (TNM) staging [57]. Ephrin-B2 seems to possess a predictive capacity for patients with a PDAC prognosis who respond to therapy [65]. Analogously, Lu et al. demonstrated that the overexpression of EPHB2 and ephrin-B2 clinically correlated with more aggressive PDAC behavior, as well as with abdominal and back pain [66]. In a recent phosphoproteomics analysis, which mirrored the activation of a multitude of signaling pathways, Renuse et al. reported the existence of 709 proteins with, overall, 1199 loci. EPHB4 in parallel with EPHA2 were identified as the molecules with the most kinase-regulating sites among the EPH/ephrin system [55]. The above points are summarized in Table 1. 

Despite the fact that the above findings may not be urgently transferable to clinical practice, several clinical trials in humans have begun to emerge [64,70,71]. To date, EPHA2 has been the only targeted molecule. Shitara et al. utilized DS-8895a, a humanized IgG1 EPHA2-targeting antibody with the capacity to augment antibody-dependent cellular cytotoxicity, in a phase I study. They demonstrated its safety at the doses utilized, as well as the activation of NK cells [70]. Regardless of the limited drug uptake from normal tissue, further clinical studies did not succeed due to poor biodistribution results [64]. Two more Phase I clinical studies including PDAC patients are currently recruiting. Weston et al. are currently investigating the effects of siRNA-EPHA2 in tumor metabolism and perfusion, utilizing diffusion weighted MRI and 18FDG-PET [72], while Huang et al. explored the efficacy of EPHA2-specific taxane-loaded immunoliposomes [71]. 

## 3. Discussion

Based on the existing literature, it is evident that the EPH/ephrin signaling system mediates a multitude of diverse physiologic processes, such as the development of placenta [73], the perception of pain [74], neurodegeneration [75] and fibrosis [76], while its impact in cancer has been extensively reviewed [21,23,33,77]. In fact, in carcinogenesis, the EPH/ephrin system mediates several pro-tumorigenic processes that comprise hallmarks of neoplasia [78], such as the interplay with proliferating signaling [79], the promotion of invasion and metastasis [80] and the induction of angiogenesis [81]. As regards the influence of the EPH/ephrin system in the modulation of immune infiltrates in TME, our knowledge has been widely extended recently [6]. Data indicate that the EPH/ephrin system exerts its regulatory effects on immune cellular populations [48,82], with a profound impact on the patient’s prognosis. 

Despite the immense scientific efforts towards deciphering PDAC’s TME [83], PDAC comprises the seventh most common cause of malignancy-related death. Recent translational data from human PDAC tissue suggested that the upregulation of 41BB and lymphocyte activation gene-3 (LAG-3) expression [84] generates a potent immunosuppressive environment. The 41BB receptor and its ligand are expressed upon T cell stimulation [85], and LAG-3 is a T-cell inhibitory co-receptor which could be exploited therapeutically [86]. Clinical studies suggested that the combinational triple targeting with 41BB agonists, LAG3 antagonists and CXCR1/2 inhibitors could result in a clinical response [84]. Additionally, the dense fibrotic stroma generates an immunosuppressive, tumor-promoting microenvironment, causing further impairment in drug delivery [87]. IL-35 constitutes a major modulator of the immune response towards PDAC, shaping both T- and B-cellular reactions. Regarding the T-cells, IL-35 is implicated in the downregulation of the effector CD4 cells and the concomitant extension of the Tregs [88]. Analogously, naive B-cells are inclined towards the Bregs over the plasma cellular phenotype under the influence of IL-35/STAT3 signaling, and the inhibition of IL-35/BCL6 signaling potentiates the immunotherapy effects [89]. The above highlight the significance of B-cells in PDAC carcinogenesis and response to immunotherapy, as knowledge of its effects has recently begun to emerge [90,91]. Although the immune system exerts fundamental effects during carcinogenesis in several malignancies [92,93,94], heretofore, the clinical trials regarding immunotherapy in PDAC have been discouraging, indicating that further work needs to be conducted [95]. Moving towards this direction, Markosyan et al. demonstrated that the therapeutic manipulation of EPHA2 could offer a valuable alternative to enhance the tumor’s responsiveness to immunotherapy [48]. In parallel, the EPHB4/ephrin-B2 targeting could suppress the pro-tumorigenic immune infiltrates, causing further tumor regression [59]. Additional research needs to be conducted in order to better unfold the complex mechanisms that mediate the aforementioned functions.

Irrespective of the great efforts that have been conducted, several drawbacks limit the transfer of the aforementioned research into clinical practice. Regardless of the fact that that there have been increasing amounts of research on PDAC, the majority of these studies are pre-clinical. It is easily perceived that large, prospective clinical trials are necessary in order to validate the effects of the EPH/ephrin signaling on humans. To date, the existing clinical studies are of limited clinical efficiency, and additional validation in larger clinical trials should be achieved in order to gain clinical significance.

## 4. Conclusions

In conclusion, the EPH/ephrin system seems to regulate embryologic development, as well as several crucial physiologic processes of the pancreas, such as the secretion of insulin and glucagon. It could, potentially, be exploited therapeutically in order to target PDAC. In fact, its implementation into clinical practice could revolutionize the management of PDAC, introducing a more efficacious immunotherapy therapeutic scheme.

## Figures and Tables

**Figure 1 ijms-24-03015-f001:**
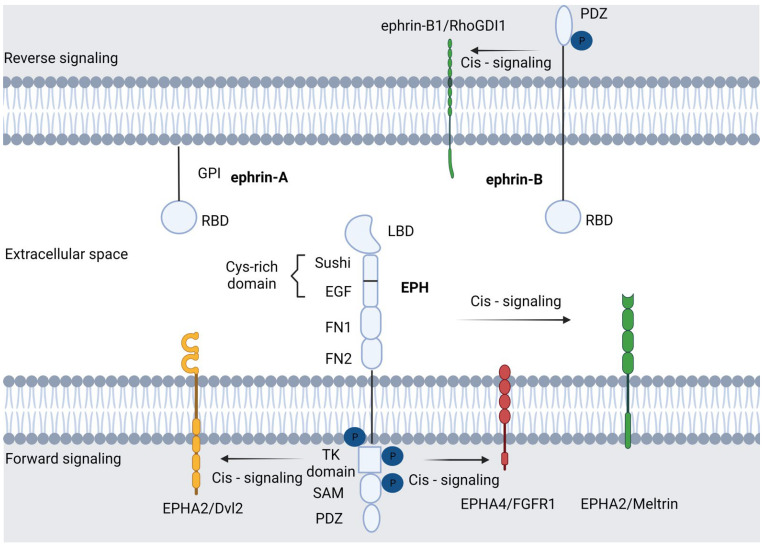
Basic molecular structure of the EPH/ephrin signaling compartment. In the plasma membrane, ephrin-A ligands are hooked by a glycosylphosphatidylinositol (GPI) anchor, although they can also activate distant EPH receptors. On the other hand, ephrin-B ligands contain a transmembrane domain and a cytoplasmic segment. EPHs/ephrin interaction results in the beginning of a molecular series of events. Forward signaling is triggered by EPH–ephrin binding as well as through EPH interplay with various biomolecules and signaling pathways. Reverse signaling is stimulated by the EPHs in the ephrin-expressing cells. Cis-signaling is the interaction of EPHs and ephrins with molecules on the same cell membrane. Proteins that encompass Src Homology 2 (SH2) or PDZ domains participate in the transmission of the downstream signal through their interaction with ephrins. Created with BioRender.com.

**Figure 2 ijms-24-03015-f002:**
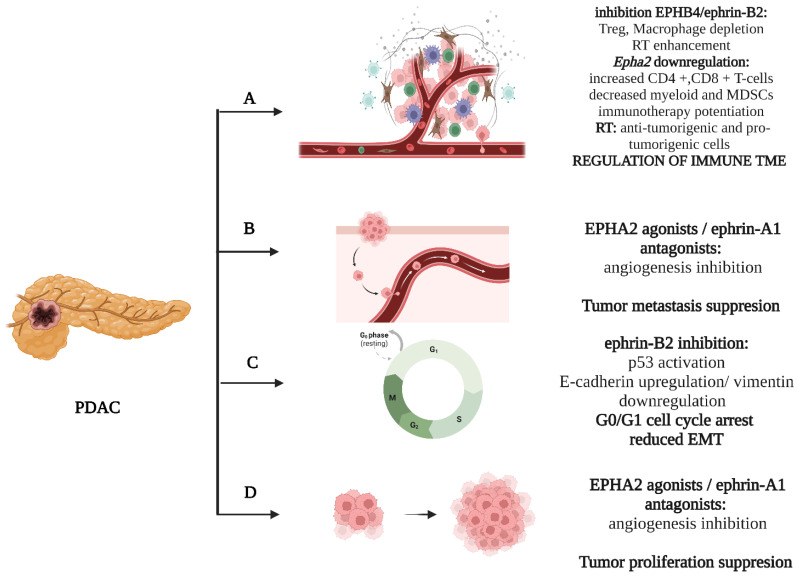
Basic mechanisms influenced by the EPH/ephrin system: (**A**) RT, EPHA2 and EPHB4/ephrin-B2 shape the immune cellular population of TME. (**B**) EPHA2/ephrin-A1 influences the metastatic potential of cancer cells. (**C**) Ephrin-B2 regulates G0/G1 cell cycle transition. (**D**) EPHA2/ephrin-A1 induce tumor growth. Created with BioRender.com.

**Table 1 ijms-24-03015-t001:** Published clinical data regarding the EPH/ephrin signaling system.

EPH/Ephrin	Study Material	Result	References
EPHA1/A2/A4/A5/A7	Neoplastic tissue	EPHA1 staining intensity was significantly associated with Ki67 expression pathologic staging	[60]
EPHA2	Neoplastic tissue	EPHA2 was associated with poor outcome and aggressive disease	[54,61]
EPHA2	Soluble EPHA2 fragments	May be applicable as a diagnostic biomarker	[62]
EPHA2	Neoplastic tissue	The expression of EPHA2 was inversely correlated with the degree of T cell infiltration in PDAC	[48]
EPHA2	PC patients	Dasatinib (inhibition of EPHA2) did not show clinical activity in metastatic PDAC	[67]
EPHA4	Neoplastic tissue	EPHA4 positivity was associated with lower overall survival	[68]
EPHB2/ephrin-B2	Neoplastic tissue	Overexpression of EPHB2 and ephrin-B2 was associated with: histologic differentiation pathologic TNM	[66]
ephrin-B2	Neoplastic tissue	High ephrin-B2 expression correlated with: poor survival in PDAC (median OS 15.80 vs. 22.83 months)	[65]
ephrin-B2	Neoplastic tissue	Lower expression of *ephrin-B2* and *ADAM10* after neo-adjuvant therapy was associated with better: overall survival (OS) disease-free survival (DFS)	[69]
EPHB4	PDAC patients	Significant expression of EPHB4 in >70% of patients with PDAC	[50]

## Data Availability

Not applicable.

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
