# Peer review of "The EPH/Ephrin System in Pancreatic Ductal Adenocarcinoma (PDAC): From Pathogenesis to Treatment"

_ijms, 2023, doi:10.3390/ijms24033015_

Round 1

Reviewer 1 Report

This review paper is highlighting literature evidence implicating the Eph/Ephrin system in pancreatic ductal adenocarcinoma. The authors have provided a concise review showcasing papers covering preclinical and translational clinical data supporting the involvement of Ephrin signaling in PDAC.

Overall, my enthusiasm for this manuscript is reduced. It is poorly written and needs major English language editing. Figure 1 has been extensively covered in other review articles and adds little to this paper. The authors should include a schematic illustrating the molecular underpinning of Ephrin signaling in PDAC and how targeting either forward or reverse signaling could potentially improve outcomes.  

Author Response

We would like to thank the reviewer for his kind and helpful comments which enabled us to improve our manuscript. Please find a point-by-point answer to your comments below:

  • It is poorly written and needs major English language editing.

We thank the Reviewer for this comment. As you suggested, we performed linguistic revision. In fact, especially the first part of the manuscript was improved drastically linguistically.

  • Figure 1 has been extensively covered in other review articles and adds little to this paper. The authors should include a schematic illustrating the molecular underpinning of Ephrin signaling in PDAC and how targeting either forward or reverse signaling could potentially improve outcomes.

We thank the Reviewer for this comment. As you suggested, we have revised Figure 1 in order to be more precise in regard with forward and reverse signaling. According to your suggestion, we added in the text and in figure a more analytic presentation of cis-signaling to better summarize the mechanisms of EPH/ephrin effects. Combined with the upgraded Figure 2, which provides more information about the therapeutic targets of EPH/ephrin signaling, readers could deepen into the current EPH/ephrin therapeutic targets in PDAC.

Reviewer 2 Report

In this manuscript, Papadakos et al. summarized progresses from both laboratory and clinical research to illustrate the role of EPH/ephrin signaling system in the progression of PDAC. The authors systematically highlighted the most important functions of EPH/ephrin family members in general biology and pancreas development and then, the specific studies in pancreas cancers and clinical statistics. This review manuscript is well-structured and systematically organized to support their conclusion on the promising future of the EPH/ephrin signaling in PDAC control. The overall quality of this manuscript is high and meet the scope of IJMS. Only one minor issue is suggested to be enhanced for more comprehensive summary of this field to the readers.

Page 2 line 79, the function and mechanism of the cis-EPH/ephrin signaling is over simplified. The cis signaling is fundamentally critical in motor axon guidance and remolding. Recent experimental studies also provided evidence of this cis regulation signaling in cancers. Please provide more information about the contribution of cis signaling in cancer progression and metastasis. If possible, a scheme of the cis signaling could be added as a panel in the Figure 1.

Author Response

We would like to thank the reviewer for his kind and helpful comments which enabled us to improve our manuscript. Please find a point-by-point answer to your comments below:

  • Page 2 line 79, the function and mechanism of the cis-EPH/ephrin signaling is over simplified. The cis signaling is fundamentally critical in motor axon guidance and remolding. Recent experimental studies also provided evidence of this cis regulation signaling in cancers. Please provide more information about the contribution of cis signaling in cancer progression and metastasis. If possible, a scheme of the cis signaling could be added as a panel in the Figure 1.

We thank the Reviewer for this comment. As you suggested, we have revised Figure 1 accordingly in order to include information about cis-signaling.  We have deepened into cis-signaling in the manuscript in order to present more information about this critical aspect of the EPH/ephrin signaling.